# Is Longstanding Congenital Muscular Torticollis Provoking Pelvic Malalignment Syndrome?

**DOI:** 10.3390/children8090735

**Published:** 2021-08-26

**Authors:** Jun-il Park, Joo-Hyun Kee, Ja Young Choi, Shin-seung Yang

**Affiliations:** Department of Rehabilitation Medicine, College of Medicine, Chungnam National University, Daejeon 35015, Korea; uniwater@cnuh.co.kr (J.-i.P.); awngusla@naver.com (J.-H.K.); jaychoi3399@gmail.com (J.Y.C.)

**Keywords:** pelvic malalignment syndrome, congenital muscular torticollis, long-term follow-up, scoliosis

## Abstract

It has been reported that congenital muscular torticollis (CMT) may result in secondary scoliosis over long-term follow-ups. However, there are few reports on whether CMT causes pelvic malalignment syndrome (PMS). This study aimed to investigate the relationship between CMT and PMS and to determine the factors associated with the development of PMS in children with longstanding CMT. Medical records of 130 children with CMT who had long-term follow-up were reviewed retrospectively. The chi-squared test and logistic regression analysis were used to determine which initial clinical parameters contributed to the development of PMS. Among 130 children with CMT, 51 (39.2%) developed PMS with or without compensatory scoliosis during long-term follow-up, indicating a high prevalence of PMS in children with a CMT history. Initial clinical symptoms such as a limited range of motion of the neck or the presence of a neck mass could not predict the development of PMS. Even if the clinical symptoms are mild, long-term follow-up of children with CMT is essential to screen for PMS.

## 1. Introduction

Congenital muscular torticollis (CMT) is a clinical symptom that commonly occurs in neonates and infants [1]. It is also known as wry neck and twisted neck, and in the case of mass-type CMT, it is called fibromatosis colli [2,3,4,5]. Shortening of the unilateral sternocleidomastoid (SCM) muscle is characteristic of the disease, and it is known to occur in 0.3–2.0% of newborns [6,7,8]. Contracture of the SCM muscle causes the face to rotate to the opposite side and the head to tilt to the same side, resulting in signs and symptoms such as a limited range of motion (LOM) in the neck [9]. Without appropriate treatment, long-term twisting of the neck may result in compensatory changes in the adjacent skeletal structures, resulting in bilateral facial asymmetry and secondary scoliosis [9,10,11,12].

The etiology of CMT remains unclear. However, previous studies have assumed that congenital trauma, perinatal compartment syndrome, and impairment of the developing SCM owing to intrauterine constraints are the main causes [13]. Additionally, secondary changes in the infant’s spine have been reported [14,15,16]. Recent studies have shown that the severity of spinal deformities in infants with CMT increases with age and with the severity of SCM tightening [13,14].

Pelvic malalignment syndrome (PMS) is a biomechanical change that might cause pain when the pelvis supporting the central axis of the body is twisted, and the entire body is out of alignment [17]. Rotational malalignment, upslip of the sacroiliac joint, and inward or outward movement are common, and if growth continues without modification, the spine and pelvis are deformed [17,18]. In severe cases, the disk may be pushed out to one side, causing spinal diseases that exert pressure on the surrounding nerves. It was reported that there was a positive correlation between the severity of scoliosis and pelvic obliquity in patients with adolescent idiopathic scoliosis. There were some children seen with both secondary scoliosis and pelvic asymmetry among the children with a history of CMT [19].

Although secondary scoliosis frequently occurs in patients with CMT, the relationship between CMT and PMS has not been evaluated. To the best of our knowledge, no study regarding the development of PMS through long-term follow-up of children with CMT exists.

Therefore, our study aimed to investigate the relationship between CMT and PMS and to determine the factors associated with the development of PMS in children with longstanding CMT.

## 2. Materials and Methods

### 2.1. Study Design

This retrospective study was conducted at a single tertiary medical center. We reviewed the medical records of children who were diagnosed with CMT between January 2011 and June 2017. This study was approved by the institutional review board of the National University Hospital (IRB no. 2017-08-008-001).

### 2.2. Subjects

We reviewed the medical records of 130 children (80 boys and 50 girls) who visited our outpatient clinic of the rehabilitation department between January 2011 and June 2017 (Table 1). The inclusion criteria were children who were diagnosed with CMT for the first time before the age of 4 years and had been followed up for more than 2 years. The included children had visited the hospital with the chief complaint of abnormal posture of the head and were diagnosed with CMT by ultrasonography and clinical examination. We enrolled both mass-type and non-mass-type CMT cases. In most cases of mass type, the mass had disappeared before 12 months, and that was confirmed by clinical examination and ultrasonography during the routine follow-up examination. Babies under 1 year were routinely followed up every 2 months. The exclusion criteria were as follows: (a) congenital anomalies of the spine, (b) spasmodic torticollis, (c) premorbid or comorbid musculoskeletal problems that affect the cervical range of motion, and (d) other congenital musculoskeletal anomalies of a lower extremity, such as leg length discrepancy or hip dysplasia. In addition, children who underwent surgical treatment for the remaining fibrotic mass were excluded from this study.

Almost all children diagnosed with CMT received regular physical therapy, and their parents were educated to do stretching of affected SCM muscles daily, even if the patients did not get the regular physical therapy.

### 2.3. Outcome Measurements

Children presenting with observable neck/head/facial asymmetry were screened for a diagnosis of CMT. Diagnosis of congenital muscular torticollis was usually based on Ballock’s algorithm [8]. The birth history was obtained by the caregiver’s interview.

Physical examinations were performed at the first visit and the regular 2-month-interval follow-ups during the first year of life and then were executed annually.

Physical examinations for the neck included SCM tightness, the presence of a fibrotic mass, the presence of plagiocephaly, and passive neck range of motion (ROM). At every hospital visit, the degree of SCM tightness was checked through stretching of each sternal belly and cleido-belly, and the severity of plagiocephaly was assessed using a diagonal caliper, which measured the difference between the diagonal lengths. Passive neck ROM was measured on both sides using a goniometer in the transverse and coronal planes (Figure 1) during the first year of life.

If there was a difference in the ROM of the SCM between the affected side and the unaffected side in either plane, we marked the case to have the LOM of the neck.

Ultrasonography for the bilateral SCM was performed by a radiologist using a commercial ultrasound system (Virtual Touch Imaging, ACUSON S2000 Ultrasound Unit, Siemens, Mountain View, CA, USA) initially. During the examination, children were placed in the supine position, with their heads slightly tilted to the opposite side and their necks gently extended by placing a small rolled towel below the shoulder. The thickness of the SCM muscle was measured in the longitudinal and transverse planes (Figure 2).

According to the standard practice guideline for CMT of our institution, routine supine cervical spine radiographs were taken at the first visit with the WHO manual of diagnostic imaging [21]. An initial cervical spine radiograph was taken for ruling out the congenital spine anomaly. Follow-up whole-spine radiographs were taken annually or every 2 years, depending on the severity of symptoms.

Whole-spine anterior-posterior (AP) radiographs were taken in the erect position of the patients. Each patient was asked to stand with both arms placed by the sides, and knees and hips extended, and with equal weight bearing on both feet. The technician ensured the patient aligned centrally to the image receptor and reduced the rotation of hips and shoulders as much as possible (Figure 3).

The physical examination for symmetry of the pelvic bone in the transverse plane and the coronal plane was executed annually after the children started walking.

For the detection of asymmetry of the pelvic bone in the coronal plane, we compared the highest point of the bilateral iliac crest in a standing position. At the same time, we compared the position of the anterior superior iliac spine (ASIS) with palpation and checked the asymmetry of rotation of the pelvis in a comfortable standing position (Figure 4).

If the pelvis aligned unequally, the actual leg length was measured with a tape measure from the ASIS to the tip of the tibial medial malleolus. We excluded cases with actual leg length discrepancy.

PMS was defined as pelvic obliquity and pelvic rotation with or without compensatory scoliosis, which was confirmed through radiologic and clinical evaluations. The diagnosis was based on the findings of asymmetrical alignment of the pelvic bone, compensatory curvatures of the spine, with or without associated malrotation of one or more vertebrae on the radiograph, and asymmetrical position and ROM of the pelvis in the transverse and coronal planes at clinical examination [17,18] (Figure 5).

### 2.4. Statistical Analysis

All data were analyzed using SPSS statistical software (version 22.0; SPSS Inc., Chicago, IL, USA) for Windows. The chi-squared test was used to determine which initial parameters contributed to the development of PMS with compensatory scoliosis. Percentages of the two groups were calculated. Logistic regression analysis was performed to identify factors affecting the development of PMS. Statistical significance was set at a *p*-value of <0.05.

## 3. Results

This study included 130 children with confirmed CMT who had a prolonged follow-up of more than 2 years. Participants’ baseline characteristics are presented in Table 1. There were 80 (61.5%) boys and 50 (38.5%) girls. In terms of the delivery type, the number of births through cesarean section was 38 (29.2%), and the number of vaginal deliveries was 92 (70.8%). There were 45 (34.6%) children with a palpable mass confirmed through physical examination and ultrasonography. There were 73 (56.2%) children with limited ROM of the neck and 34 (26.2%) children with plagiocephaly.

Surprisingly, PMS with compensatory scoliosis or without compensatory scoliosis was observed in 51 (39.2%) of 130 children diagnosed with CMT during the long follow-up period of more than 2 years. There were 20 (15.4%) children who developed scoliosis and 48 (36.9%) children who developed only PMS, while 3 (2.3%) children developed the PMS combined with scoliosis (Table 2). The changes in the spine and pelvic alignment according to age at follow-up are presented in Table 3. Although not all children had radiographs taken at the same timeline, children who visited the clinic before 12 months of age began to show PMS after about 3 years generally.

There was no significant difference in the initial clinical picture between the group with PMS and the group without PMS (Table 4). Univariate logistic regression analysis revealed no significant factors associated with the development of PMS (Table 5).

## 4. Discussion

In our retrospective study, we found a high prevalence of PMS in children with a history of CMT. Among 130 children, 51 (39.2%) developed PMS. However, no significant association was found between the variable clinical factors and PMS development. Thus, it was suggested that CMT might have a clinically significant association with the development of PMS. Although initial clinical manifestations were not severe, children with CMT could develop PMS after a long time.

CMT could be classified into postural, muscular, and mass types, and Cheng et al. reported an incidence rate of 22.1%, 30.6%, and 42.7%, respectively [22,23]. Postural CMT is caused by a preference for the head posture by infants without LOM. [23]. Muscular CMT is not caused by a fibrous tumor but is caused by the tight muscle, resulting in limitation of ROM. However, in a clinical setting, it is difficult to distinguish postural CMT from muscular CMT.

Mass-type CMT called fibromatosis colli [24] has the challenge to secure the full ROM of the neck and to prevent facial asymmetry and is associated with a longer treatment period [21]. Fibromatosis colli has an incidence rate of approximately 0.3–2% of births [2]. This mass usually grows by 1–2 months of age and then gradually decreases in size and eventually disappears by the second year [25]. The progression in size is due to collagen deposition and muscular fibrosis, where fibroblasts migrate to individual muscle fibers that undergo atrophy [26]. The severity of the mass-type CMT differs depending on the degree of fibrosis. Muscular fibrosis with reduced elasticity limits the ROM of the neck by reducing muscle elasticity and can induce contraction of muscle [27]. Recent immunological studies have confirmed that the deposition of type III collagen is a key factor in the production of SCM fibrosis, and its hyperplasia is linked to accelerating apoptosis and the overexpression of growth factor beta-1 [13,28]. Gene expression studies have supported this by suggesting that the pathogenesis of CMT might be related to fibrosis, with collagen and elastin fibrillogenesis causing the mechanical strain [1,13].

In this study, we expected that children with a history of CMT with a mass would be more likely to have PMS later, but the result was not different between the two groups. It was inferred that the outcome being different from the expectation was due to the diverse clinical course of the mass type, depending on the histological characteristics of the mass.

Generally, CMT is diagnosed not on the basis of muscle biopsy but on the basis of clinical features and non-invasive ultrasonography, so the histological feature cannot be assessed precisely. Although a mass was not present initially, the reduced muscle elasticity eventually resulted in postural imbalance later. Consequently, it was judged that the presence or absence of a fibrotic mass did not seem to affect the occurrence of the PMS.

As previously mentioned, PMS can be defined as an asymmetric condition when the pelvis supporting the central axis of the body is twisted, and the entire body is out of alignment. Muscular weakness and pain can develop by repetitive movements or persistent asymmetric posture in everyday life. Asymmetric flare affects the distortion of the pelvic ring in the transverse plane and consequently provides asymmetric tension in ligaments and muscles. Additionally, slip and rotational malalignment distort the pelvic joints, resulting in differences in the pelvic slope and functional leg length, and the compensatory curvature of the spine to maintain the horizontal and central axis of the head [17]. Although these findings could increase the chances in children with CMT combined with PMS to complain of pain when doing exercise, there were no pains reported in this study.

CMT decreases muscle elasticity, resulting in differences in the active ROM of the neck. If this phenomenon is maintained for a long time without proper intervention, it can cause an asymmetric face and trunk, such as scoliosis and pelvic malalignment. To keep the eye in front, children with CMT might perform a compensatory rotation of the trunk to the direction of the affected muscle to avoid stretching of the tight sternal belly of the SCM in the transverse plane. Moreover, there could be a compensatory motion of the cervical spine and lifting of the affected side of the shoulder to keep both eyes horizontally, avoiding stretching of the tight cleido-belly of the SCM. Thus, an altered cervical spine curve might provide a compensatory effect on the whole spine and pelvis in the coronal plane. This compensatory movement for tight SCM muscles in both transverse and coronal planes might be the explainable biomechanical mechanism.

Another important finding of this study was that the change in elasticity of the affected SCM muscle has a possibility of a negative long-term effect up to the symmetry of the pelvic bone regardless of the initial severity of the symptom. The slight muscular fibrosis may persist in non-mass-type CMT so that it cannot be said that the elasticity of the affected muscle is exactly the same to prevent asymmetry of the trunk. Even though children of CMT with or without a mass undergo physical therapy earlier, they might have PMS after a long-term period.

In our study, no significant association was found between the initial clinical manifestation and the progression of PMS. In other words, even though the initial clinical symptom was not severe, the fine fibrosis of SCM muscles might result in the development of PMS.

Generally, CMT is diagnosed not on the basis of muscle biopsy but on the basis of clinical features and non-invasive ultrasonography, so the histological feature cannot be assessed precisely. Although a mass is not presented initially, the reduced muscle elasticity results in postural imbalance later. Consequently, it is judged that the presence or absence of a fibrotic mass does not affect the occurrence of the PMS.

The prevalence of secondary scoliosis and its association with CMT have been investigated in many studies [15]. In one study, the prevalence of secondary scoliosis in patients with CMT was 82.1% [9]. However, no study has reported the factors associated with the development of PMS. Our study could be the evidence that suggests the need for long-term follow-up for screening the development of PMS and scoliosis in children with CMT.

Our study had several limitations. First, all data of this study were obtained retrospectively. According to standard practice guidelines for CMT, a cervical spine radiograph was taken immediately after visiting, and then a whole-spine radiograph was taken every 1 year or 2 years as a follow-up. However, all children did not undergo radiograph examination at the same age. So, we could get the only approximate occurring time of PMS but could not get the accurate natural progression. Second, the intensity and amount of physical therapy were not controlled thoroughly. Although all participants actively participated in physical therapy, the number and time of participation were not carefully tailored. Third, the number of participants was relatively small, and the follow-up period was relatively short, which was about 3.5 years.

## 5. Conclusions

In summary, children with a history of CMT showed a high prevalence of PMS. However, many clinical parameters such as sex, age of diagnosis, lesion side, initial neck LOM, initial neck mass size, delivery type, and plagiocephaly could not predict the development of PMS. Therefore, long-term follow-ups are needed in all children with CMT regardless of the severity of initial symptoms. Further prospective studies are needed to delineate the factors associated with PMS in children with CMT.

## Figures and Tables

**Figure 1 children-08-00735-f001:**
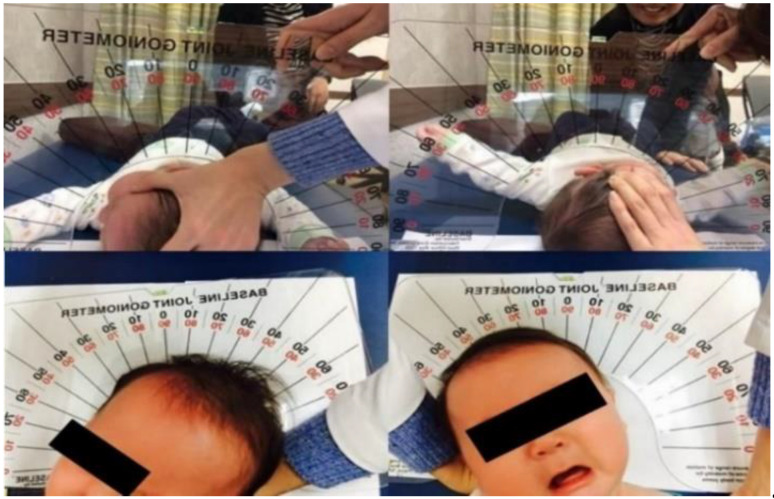
Cervical range-of-motion measurement using arthrodial goniometer (adapted from Hwang et al. Changes in muscle stiffness in infants with congenital muscular torticollis 2019; 9(4); 158 [20]).

**Figure 2 children-08-00735-f002:**
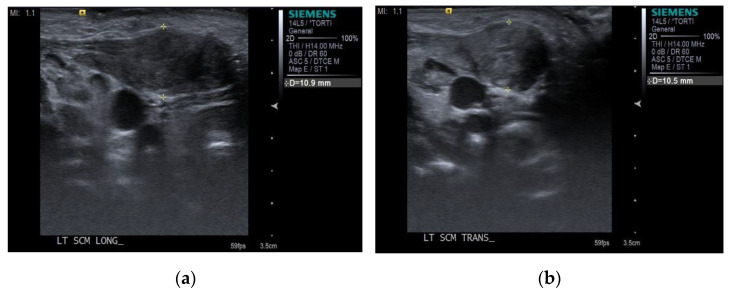
The thickness of the SCM muscle was measured with electronic calipers in (**a**) the longitudinal plane and (**b**) in the transverse plane.

**Figure 3 children-08-00735-f003:**
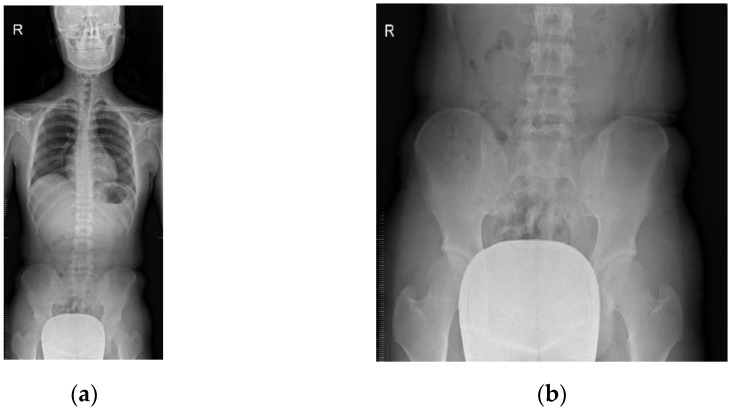
Radiographic changes with malalignment: (**a**) whole-spine AP radiograph and (**b**) pelvic portion radiograph.

**Figure 4 children-08-00735-f004:**
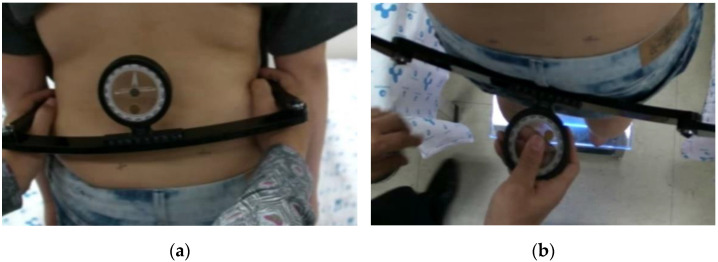
Measurement of pelvic asymmetry in (**a**) the coronal plane and (**b**) in the transverse plane.

**Figure 5 children-08-00735-f005:**
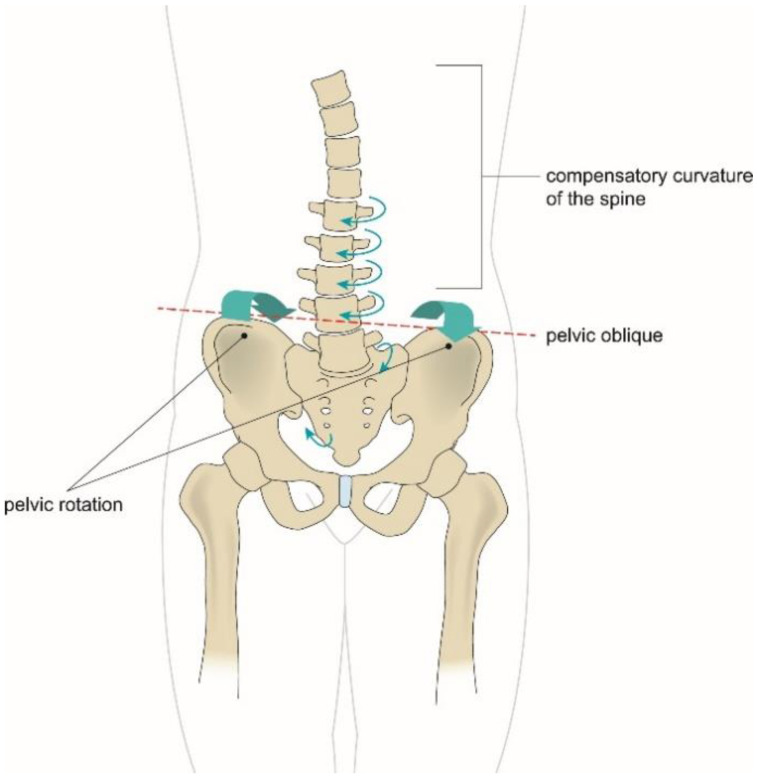
Asymmetrical alignment of the pelvic bone, compensatory curvatures of the spine, and asymmetrical ranges of motion of the pelvis in transverse and coronal planes.

**Table 1 children-08-00735-t001:** Characteristics of children with CMT (*n* = 130).

Characteristic		*n* (%)
Sex	Male	80 (61.5)
	Female	50 (38.5)
Initial diagnosis age	0–12 months	124 (95.4)
	13–24 months	3 (2.3)
	After 24 months	3 (2.3)
Delivery type	Cesarean section	38 (29.2)
	Vaginal delivery	92 (70.8)
Range of motion	Limited	73 (56.2)
	Normal	57 (43.8)
Presence of fibrotic mass	Presence	45 (34.6)
	Absence	85 (63.4)
Plagiocephaly	Presence	34 (26.2)
	Absence	96 (73.8)

Note: Values are presented as numbers (%).

**Table 2 children-08-00735-t002:** Changes in the spine and pelvic alignment after more than 2 years of follow-up among 130 children.

	*n* (%)
Scoliosis	20 (15.4)
Pelvic malalignment syndrome (PMS)	48 (36.9)
Scoliosis combined with PMS	3 (2.3)
No change	59 (45.4)
Total	130 (100.0)

Note: Values are presented as numbers (%).

**Table 3 children-08-00735-t003:** Changes in the spine and pelvic alignment according to the ages at follow-up visits.

F/U Age (*n*)	PMS	Scoliosis	Scoliosis Combined with PMS
24–48 months (77)	24 (18.5)	5 (3.8)	1 (0.8)
49–60 months (24)	12 (9.2)	4 (3.1)	0 (0.0)
61 months–12 year (29)	12 (9.2)	11 (8.5)	2 (1.5)
Total (130)	48 (36.9)	20 (15.4)	3 (2.3)

Note: Values are presented as numbers (%).

**Table 4 children-08-00735-t004:** Comparison according to the pelvis alignment status group at long-term follow-up.

Factor	Malalignment Pelvis Group (*n* = 51)	Normal Pelvis Group (*n* = 79)	*p*-Value
Delivery type			0.451
Cesarean section	13 (34.2)	25 (65.8)	
Vaginal delivery	38 (41.3)	54 (58.7)	
Location			0.590
Right	24 (36.9)	41 (63.1)	
Left	27 (41.5)	38 (58.5)	
Range of motion			0.622
Limited	30 (41.1)	43 (58.9)	
Normal	21 (36.8)	36 (63.2)	
Presence of mass			0.611
Presence	19 (42.2)	26 (57.8)	
Absence	32 (37.6)	53 (62.8)	
Plagiocephaly			0.584
Presence	12 (35.3)	22 (64.7)	
Absence	39 (40.6)	57 (59.4)	

Note: Values are presented as numbers (%).

**Table 5 children-08-00735-t005:** Logistic regression analysis of factors associated with pelvic malalignment syndrome.

	Univariate Analysis	
Variable	B (SE)	*p*-Value
Female vs. male	0.32 (0.37)	0.379
Diagnosis age, <6 months	0.31 (0.47)	0.513
Involved side, right vs. left	0.19 (0.36)	0.590
Initial neck LOM	0.23 (0.36)	0.527
Concurrent plagiocephaly	−0.28 (0.43)	0.518
Fibromatosis colli type	−0.19 (0.38)	0.611

SE, standard error; LOM, limitation of range of motion.

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
