# Peer review of "Is Longstanding Congenital Muscular Torticollis Provoking Pelvic Malalignment Syndrome?"

_children, 2021, doi:10.3390/children8090735_

Round 1
Reviewer 1 Report
This is an interesting topic to those interested in congenital muscular torticollis and potential resulting asymmetries.
Throughout the paper, there are points that are unclear. For example, how did the diagnosis of malalignment syndrome occur? Were all children given x-rays at a particular age? At what age did these findings manifest? How often were the children x-rayed? Were there complaints of pain? The authors talk about pain, structural deformities and cosmesis being potential problems but it is unclear if pain or complaints about cosmesis led to the x-rays or if they were routine.
Also, in the inclusion criteria, presence of torticollis before age 4 years would appear to include torticollis that could have occurred from a number of causes - what would cause such a late referral? Why not referrals within the first year of life?
Lines 108-109 - "When measuring the neck ROM, if there was a difference between the left and right ROMs, we decided that the ROM of the affected side was limited" - actually the range to the opposite side of the tight muscle should be affected or diminished
Line 115 - the authors use "we" - does this mean that the authors were the radiologists? Typically use of 3rd person is more common especially if the authors were not the radiologists.
Starting at line 122 - the methods in the preceding section on the US was very carefully described but the methods for the x-rays to determine if the child had malalignment syndrome were not described. How did the examiners assure that the alignment seen was not simply how the child was placed on the table if supine? Or were the x-rays performed in standing? If so, was leg length taken into account? This part is a major issue for the premise of the study and must be explained.
The paper states that the x-rays were at a "minimum of 2 year follow up" - this could make some children 6 years of age and others 2 years of age. If 2 years was the minimum, some children might be even older. This needs to be specified. Ages are never presented in this paper and need to be addressed.
In the discussion section, the authors talk about the muscular form of torticollis as if an entirely separate entity from the pseudotumor type. However, the tumor typically disappears very early so how can it be determined that the tumor did not exist at one point in time?
lines 228-229 - this part is very confusing "Secondary shortening in children with CMT can be observed in the convex trunk muscles on the same or opposite sides" - why would there be secondary shortening on the convex side? Why would that side not be lengthened?
There are several places in the paper that require revision for English grammar.
Lines 34-35 - "rotate on opposite sides" should be "rotate to the opposite side"
Line 57 - "it is important to not miss appropriate calibration time through early screening" - "to not miss" is awkward wording...and "appropriate calibration time" is confusing
Lines 90-91 - "screened for diagnose CMT" is grammatically incorrect - should perhaps be "screened for a diagnosis of CMT" or similar
Line 96 is missing a space between "for" and "SCM"
Line 219 - " Off-normal damage" needs clarification - what does this mean?
Author Response
We have sincerely appreciated your comments on improving the quality of our paper. We have addressed all the comments and made a major revision of our manuscript. Our responses to all your comments and the revisions are listed below.
This is an interesting topic to those interested in congenital muscular torticollis and potential resulting asymmetries. Throughout the paper, there are points that are unclear.
For example, how did the diagnosis of malalignment syndrome occur?
Response: PMS was diagnosed by radiologic evaluation of the spine X-ray radiographs and clinical evaluation. The diagnosis was based on the findings of 1) asymmetrical alignment of the pelvic bone, 2) compensatory curvatures of the spine, with or without associated malrotation of one or more vertebrae, and 3) asymmetrical ranges of motion of the pelvis in the transverse and sagittal planes.
[Reference]
- Schamberger, W. Malalignment Syndrome in Runners. Phys Med Rehabil Clin N Am 2016, 27, 237–317.
doi:10.1016/j.pmr.2015.08.005.
- Schamberger, W. The Malalignment Syndrome, 2nd Ed; Churchill Livingstone, Canada, 2012
We added a more detailed explanation for PMS diagnosis at method and discussion.
[Added paragraph at Method] Line. 157-172
The physical examination for symmetry of pelvic bone in the transverse plane and sagittal plane was executed annually after the children started walking.
For the detection of asymmetry of the pelvic bone in the sagittal plane, we compared the highest point of the bilateral iliac crest with the pelvic leveling at a standing position. At the same time, we compared the position of the anterior superior iliac spine (ASIS) with the palpation and checked the asymmetry of rotation of the pelvis at a comfortable standing position (Figure 4).
If the pelvis leveled unequally, the actual leg length was measured with a tape measure from ASIS to the tip of tibial medial malleolus. We excluded the cases with actual leg length discrepancy.
[Added paragraph at Discussion] Line. 286-296
CMT decreased muscle elasticity, resulting in differences in the active ROM of the neck. If this phenomenon is maintained for a long time without proper intervention, it can cause an asymmetric face and trunk such as scoliosis and pelvic malalignment. To keep on eye front, the children with CMT might do a compensatory rotation of the trunk to the direction of the affected muscle to avoid the stretching of the tight sternal belly of SCM in the transverse plane. Moreover, there could be a compensatory motion of the cervical spine and lifting of the affected side of the shoulder to keep both eyes horizontally avoiding stretching of tight cleido-belly of SCM. Thus, an altered cervical spine curve might provide a compensatory effect on the whole spine and pelvis in the sagittal plane. This compensatory movement for tight SCM muscles in both the transverse and sagittal planes could be the explainable biomechanical mechanism.
Were all children given x-rays at a particular age?
Response: Our retrospective study enrolled the children whose ages ranged from 0 to 4 years old at initial diagnosis. According to standard practice guidelines for CMT, a cervical spine radiograph was taken immediately after visiting, and then every one year or two-year as a follow-up. However, not all children were equally taken radiographs at a particular age.
We added this point as the limitation of this study in the discussion part.
[Added paragraph at Discussion] Line. 322-325
According to standard practice guidelines for CMT, cervical spine radiograph was taken immediately after visiting, and then every one year or two-year as a follow-up. However, not all children were equally taken radiographs at a particular age. So, we could get the only approximate occurring time of PMS but could not get the accurate natural progression.
At what age did these findings manifest?
Response: Although not all children were taken x-rays at the same particular time, children diagnosed under 1 year of age began to show PMS from the initial diagnosis generally. Some cases over 4 years old who were referred lately as the neglected CMT already showed the pelvic asymmetry of PMS.
Also, we added the fact that we did not take a follow-up x-ray at the same particular time as an imitation of our study in our manuscript.
[Added paragraph at Discussion] Line. 321-330
Our study had several limitations. First, all data of this study were obtained retrospectively. According to standard practice guidelines for CMT, a cervical spine radiograph was taken immediately after visiting, and then a whole spine radiograph was taken every one year or two-year as a follow-up. However, all children did not undergo radiograph examination at the same age. So, we could get the only approximate occurring time of PMS but could not get the accurate natural progression. Second, the intensity and amount of physical therapy were not controlled thoroughly. Although all participants actively participated in physical therapy, the number and time of participation were not carefully tailored. Third, the number of participants was relatively small, and the follow-up period was relatively short which was about 3.5 years.
How often were the children x-rayed?
Response: All included children with CMT have been regularly examined at the outpatient clinic of the rehabilitation medicine department. According to the standard practice guideline of our institution for CMT, X-rays were taken at the time of the visit and subsequently every one or two years depending on the severity of the symptoms.
We added more detail of the X-ray schedule at Method.
[Added paragraph at Method] Line. 141-145
According to the standard practice guideline for CMT of our institution, the routine supine cervical spine radiographs were taken at first visiting with WHO manual of diagnostic imaging [20]. Taking initial cervical spine radiograph was for ruling out the congenital spine anomaly. Follow-up whole spine radiographs were taken annually or every two years depending on the severity of symptoms.
Were there complaints of pain?
Response: All information including complaints of pain has been gathered from the medical records. We researched the medical records of 130 children again, and there were no records related to the pain. The reason we mentioned the pain in our manuscript was that the children with PMS were more likely to complain the pain at doing exercise.
The authors talk about pain, structural deformities and cosmesis being potential problems but it is unclear if pain or complaints about cosmesis led to the x-rays or if they were routine.
Response: According to standard practice guidelines for CMT of our institution, the routine spine x-ray was taken at the first visit. Taking initial spine X-rays is for ruling out the congenital spine anomaly. Routine follow-up whole spine X-rays are taken annually or every two years depending on the severity of symptoms. The children with CMT had visited the clinic with the chief complaint of the abnormal posture of head. Our team had the many clinical data about children with muscular torticollis and found that they had the gradual asymmetry of facial and spine curvature after a long period even if they had a mild initial symptom. Based on these clinical experiences, our institute has made a such clinical guideline for children with CMT.
Also, in the inclusion criteria, the presence of torticollis before age 4 years would appear to include torticollis that could have occurred from a number of causes - what would cause such a late referral? Why not referrals within the first year of life?
Response: Based on your comments, we reviewed the data about the time of diagnosis again and presented the results in Table 1. Most of the cases (95.4%) visited the clinic before 12 months, and only 6 cases (4.6%) were referred after 12 months. There were some cases referred lately in which the asymmetry of the face and the trunk posture gradually worsened through the follow-up of the local clinic. We assumed that the parents could not notice the symptom of the baby because the initial symptom was not severe in some cases.
Lines 108-109 - "When measuring the neck ROM, if there was a difference between the left and right ROMs, we decided that the ROM of the affected side was limited"
- actually, the range to the opposite side of the tight muscle should be affected or diminished.
Response: In our study, we measured the passive range of motion of SCM muscles using a goniometer in transverse and sagittal planes. If the affected SCM muscle is tight as your comment, the head is tilted to the affected side in the sagittal plane and is rotated to the unaffected side in the transverse plane. If there was a difference in range of motion of SCM between the affected side and unaffected side in either one plan, we marked the case to have the limited range of motion of the neck (LOM).
A grammatical error in our manuscript has given you some misunderstanding and so that, we corrected those sentences as below.
[Added paragraph at Method] Line. 126-127
If there was a difference in ROM of SCM between the affected side and unaffected side in either one plan, we marked the case to have the LOM of the neck.
Line 115 - the authors use "we" - does this mean that the authors were the radiologists? Typically use of 3rd person is more common especially if the authors were not the radiologists.
Response: Thank you for your comment. We corrected the grammatical error in our manuscript.
Starting at line 122 - the methods in the preceding section on the US were very carefully described but the methods for the x-rays to determine if the child had malalignment syndrome were not described.
How did the examiners assure that the alignment seen was not simply how the child was placed on the table if supine?
Response: Thank you for your comments. Radiologic findings are an important part of PMS diagnosis, but it seems that there was not enough explanation for that as you mentioned. As your valuable comments, we added a detailed description of spine radiographs in our manuscript.
[Added paragraph at Method] Line. 141-151
According to the standard practice guideline for CMT of our institution, the routine supine cervical spine radiographs were taken at first visiting with WHO manual of diagnostic imaging [20]. Taking an initial cervical spine radiograph was for ruling out the congenital spine anomaly. Follow-up whole spine radiographs were taken annually or every two years depending on the severity of symptoms.
Whole spine anterior-posterior (AP) radiographs were taken in the erect position of patients. The patients were asked to stand with both arms placed by sides, and knees and hips extended, and with equal weight bearing on both feet. The technician ensured the patient aligned centrally to the image receptor, and reduced the rotation of hips and shoulders as much as possible (Figure 3).
Or were the x-rays performed in standing? If so, was leg length taken into account? This part is a major issue for the premise of the study and must be explained.
Response: We are very appreciative of your valuable comments and important pointing out.
We only included the children with only CMT initially and excluded the cases with other musculoskeletal problems including the leg length discrepancy. If the pelvis leveled unequally during the examination of the pelvic bone, the actual leg length was measured with a tape measure from ASIS to the tip of tibial medial malleolus. Also, when we rechecked the leg length discrepancy in all available radiographic images (103 cases) and medical records (30 cases), there was no such case. The orthoroentgenogram was taken with spine radiogrphs in most cases but not all (103 cases).
We added more explanation about physical examination for pelvic symmetry in our manuscript.
[Added paragraph at Method] Line. 157-172
The physical examination for symmetry of pelvic bone in the transverse plane and sagittal plane was executed annually after the children started walking.
For the detection of asymmetry of the pelvic bone in the sagittal plane, we compared the highest point of the bilateral iliac crest with the pelvic leveling at a standing position. At the same time, we compared the position of the anterior superior iliac spine (ASIS) with the palpation and checked the asymmetry of rotation of the pelvis at a comfortable standing position (Figure 4).
If the pelvis leveled unequally, the actual leg length was measured with a tape measure from ASIS to the tip of tibial medial malleolus. We excluded the cases with actual leg length discrepancy.
The paper states that the x-rays were at a "minimum of 2 year follow up" - this could make some children 6 years of age and others 2 years of age. If 2 years was the minimum, some children might be even older. This needs to be specified. Ages are never presented in this paper and need to be addressed.
Response: Our study as a retrospective study enrolled the children with CMT who were visited the clinic before the age of four. According to the standard practice guideline of our institution for CMT, X-rays were taken at the time of the visit and subsequently every one or two years depending on the severity of the symptoms.
We added the age at the initial diagnosis of the enrolled patient at Table1.
Table 1. Characteristics of patients with CMT (N = 130)
Characteristic |
n (%) |
|||
Sex |
Male |
80 (61.5) |
||
Female |
50 (38.5) |
|||
Age at initial diagnosis for CMT |
0-12 months |
124 (95.4) |
||
13-24 months |
3 (2.3) |
|||
After 24 months |
3 (2.3) |
|||
Delivery type |
Cesarean section |
38 (29.2) |
||
Vaginal delivery |
92 (70.8) |
|||
Range of motion |
Limited |
73 (56.2) |
||
Normal |
57 (43.8) |
|||
Presence of |
Presence |
45 (34.6) |
||
fibrotic mass |
Absence |
85 (63.4) |
||
Plagiocephaly |
Presence |
34 (26.2) |
||
Absence |
96 (73.8) |
|||
We also added data about the development of PMS or scoliosis or both according to the age of follow-up visit at Table 3.
Table 3. Changes of the spine and pelvic alignment according to the ages of a follow-up visit
F/U age (n) |
PMS |
Scoliosis |
Scoliosis combined with PMS |
24-48months (77) |
12 (18.5) |
5 (3.8) |
1 (0.8) |
49-60months (24) |
12 (9.2) |
4 (3.1) |
0 (0.0) |
61months-12yr (29) |
14 (9.2) |
11 (8.5) |
2 (1.5) |
Total (130) |
48 (36.9) |
20 (15.4) |
3 (2.3) |
In the discussion section, the authors talk about the muscular form of torticollis as if an entirely separate entity from the pseudotumor type. However, the tumor typically disappears very early so how can it be determined that the tumor did not exist at one point in time?
Response: In this study, we enrolled the cases with CMT of both tumor types and pseudotumor (table 1) In most cases of tumor type CMT, the mass had disappeared before 12 months, and that was confirmed by clinical examinations and ultrasonography during the routine follow-up examination (Babies under one-year-old are routinely followed up every 2 months.). Also, children who got the surgical treatment for remaining the fibrotic mass were excluded from this study. The important finding of this study is that the change of elasticity of affected SCM muscle had a possibility of a long-term effect on the symmetry of pelvic bone regardless of the initial severity of the symptom.
We added the same sentences in the Method and discussion part.
[Added paragraph at Method] Line. 79-92
We reviewed the medical records of 130 children (80 boys and 50 girls) who visited our outpatient clinic of the rehabilitation department between January 2011 and June 2017 (Table 1). The inclusion criteria were the children who were diagnosed with CMT for the first time before the age of 4 years and had been followed up for more than 2 years. The included children had visited the hospital with the chief complaint of the abnormal posture of the head and were diagnosed with CMT by ultrasonography and clinical examination. We enrolled both mass type and non-mass type of CMT. In most cases of mass type, the mass had disappeared before 12 months, and that was confirmed by clinical examination and ultrasonography during the routine follow-up examination. Babies under one year were routinely followed up every 2 months. The exclusion criteria were as follows: (a) congenital anomalies of the cervical spine, (b) spasmodic torticollis, and (c) premorbid or comorbid musculoskeletal problems that affect the cervical range of motion. Also, children who got the surgical treatment for remaining the fibrotic mass were excluded from this study.
[Added paragraph at Discussion] Line. 298-300
Another important finding of this study is that the change of elasticity of affected SCM muscle had a possibility of a long-term effect on the symmetry of pelvic bone regardless of the initial severity of the symptom.
lines 228-229 - this part is very confusing "Secondary shortening in children with CMT can be observed in the convex trunk muscles on the same or opposite sides" - why would there be secondary shortening on the convex side? Why would that side not be lengthened?
Response: The main point of that sentence was trying to say that the SCM muscle of CMT had the decreased elasticity resulting in the different active range of motion of the neck. If this phenomenon is maintained for a long time without proper intervention, it can cause an asymmetric face and trunk such as scoliosis and pelvic malalignment. It seems to give you some misunderstanding due to incorrect word selection of words. After deleting the sentence that you pointed out, we added the sentence in detail to make it easier to understand.
[Added paragraph at Discussion] Line. 286-296
CMT decreased muscle elasticity, resulting in differences in the active ROM of the neck. If this phenomenon is maintained for a long time without proper intervention, it can cause an asymmetric face and trunk such as scoliosis and pelvic malalignment. To keep on eye front, the children with CMT might do a compensatory rotation of the trunk to the direction of affected muscle to avoid the stretching of the tight sternal belly of SCM in the transverse plane. Moreover, there could be a compensatory motion of the cervical spine and lifting of the affected side of the shoulder to keep both eyes horizontally avoiding stretching of tight cleido-belly of SCM. Thus, an altered cervical spine curve might provide a compensatory effect on the whole spine and pelvis in the sagittal plane. This compensatory movement for tight SCM muscles in both the transverse and sagittal planes could be the explainable biomechanical mechanism.
There are several places in the paper that require revision for English grammar.
Response: We appreciate your advice. To improve readability, we reviewed and corrected the wrong grammar errors once again.
Lines 34-35 - "rotate on opposite sides" should be "rotate to the opposite side"
Response: Thank you for your comment. We corrected it.
Line 57 - "it is important to not miss appropriate calibration time through early screening" - "to not miss" is awkward wording...and "appropriate calibration time" is confusing
Response: Thank you for your comment. We corrected it.
Line.55-56 Therefore, it is important to check the symmetry of trunk during rapid growth of children with risk factor of asymmetrical trunk.
Lines 90-91 - "screened for diagnose CMT" is grammatically incorrect - should perhaps be "screened for a diagnosis of CMT" or similar
Response: Thank you for your comment. We corrected it.
Line 96 is missing a space between "for" and "SCM"
Response: Thank you for your comment. We corrected it.
Line 219 - " Off-normal damage" needs clarification - what does this mean?
Response: Thank you for your comment. It was misused them.

Reviewer 2 Report
Congratulations for the work done. The authors have reported a very good description of the relationship between CMT and PMS as well as factors related to the development of PMS in children with longstanding CMT. I think it has important scientific and social value.
However, I think it should improve in some aspects.
Title
The title does not inform the purpose of the study that is carried out.
Abstract
Page 1. Line. 15. (PMS) Please include the full terms before using abbreviation.
Material and Mehotds
Page. 2. Line. 79. (Table 1) Tables should be inserted into the main text close to their first citation.
Page. 2. Line. 95. Please use a period at the end of the sentence.
Page. 3. Line. 96. Please check the space between words.
Page. 3. Line. 96-98. In the section on physical examination, the measurement methods for passive neck ROM and thickness of SCM muscles are well described. A description briefly of the other physical examinations mentioned in the manuscript, such as SCM tightness, facial asymmetry, location of torticollis, delivery type and plagiocephaly, would be helpful for the reader's understanding.
Page. 3. Line. 116. (... and transverse planes and described...) I recommended inserting commas for completeness of sentences.
Page. 3. Line. 117. (figure 2) Please captitzlize the first letter.
Page. 3. Line. 118. Thickness of SCM muscle measured by electronic caliper in the longitudinal plane should be provided to help readers better understand the measurement.
Page. 4. Line. 124. (AP) Please include the full terms before using abbreviation.
Page. 4. Line. 133. I think the figure in Figure 4 is too simple compared to the content mentioned in the text. It would be better to include a picture related to the diagnosis which was based on the findings.
Page. 4. Line. 142. Please check the space between words.
Results
Page. 5. Line. 150. Please put "children" after the number.
Page. 5. Line. 146. Please control the position of numbers and parentheses equally throughout the text (abstract, result, and discussion).
ex) 80 boys (61.5%) or 45 (34.6%) children
Page. 5. Line. 156. (Table 2) Tables should be inserted into the main text close to their first citation.
Page. 6. Line. 171. (Table 3) The number of patients in the normal pelvis group (n =79) and the number in the factor (delivery type, n = 81) do not match.
Discussion
It is very extensive and must be more specific. In particular, a detailed descriptions of the following two issues are required. 1) Why were there no significant differences in the results? 2) What are the important findings of this study?
The grammer of the writing has some errors throughout the manuscript and need to be checked and corrected. This will improve the work greatly and will make it more readable.
After implementation of the changes suggested in the comment above, I support publication of the reviewed manuscript.
Author Response
We have sincerely appreciated your comments on improving the quality of our paper. We have addressed all the comments and made a major revision of our manuscript. Our responses to all your comments and the revisions are listed below.
Title
The title does not inform the purpose of the study that is carried out.
Response: The most important finding in our study is that the difference in the elasticity of
SCM muscle by muscle fibrosis causes long-term changes in trunk posture. Considering this
point, we changed the title to "Is Longstanding Congenital Muscular Torticollis Provoking
Pelvic Malalignment Syndrome?”
Abstract
Page 1. Line. 15. (PMS) Please include the full terms before using an abbreviation.
Response: Sorry for the omission of the full terms. We have corrected it again.
Material and Methods
Page. 2. Line. 79. (Table 1) Tables should be inserted into the main text close to their first citation.
Response: Table 1 was relocated to the main text close to the first cation.
Page. 2. Line. 95. Please use a period at the end of the sentence.
Response: Sorry for the omission of a period at the end of the sentence. We have corrected it again.
Page. 3. Line. 96. Please check the space between words.
Response: We have corrected it again.
Page. 3. Line. 96-98. In the section on physical examination, the measurement methods for passive neck ROM and thickness of SCM muscles are well described. A description briefly of the other physical examinations mentioned in the manuscript, such as SCM tightness, facial asymmetry, location of torticollis, delivery type, and plagiocephaly, would be helpful for the reader's understanding.
Response: we have overlooked the lack of sufficient explanation for physical examination. To make it easier for the reader to understand, we added a description of the Physical examination method for each content.
[Added paragraph] Line. 113-119
Physical examinations for the neck included the SCM tightness, the presence of fibrotic mass,
the presence of plagiocephaly, and passive neck range of motion (ROM). At every hospital visit, the degree of SCM tightness was checked through stretching of each sternal-belly and cleido-belly and the severity of plagiocephaly was assessed using a diagonal caliper which measured the difference between the diagonal lengths. Passive neck ROM was measured on both sides using a goniometer in the transverse and sagittal planes (Figure 1) during the 1st year of life.
Page. 3. Line. 116. (... and transverse planes and described...) I recommended inserting commas for the completeness of sentences.
Response: We have corrected it again.
Page. 3. Line. 117. (figure 2) Please capitalize the first letter.
Response: We have corrected it again.
Page. 3. Line. 118. The thickness of SCM muscle measured by an electronic caliper in the longitudinal plane should be provided to help readers better understand the measurement.
Response: Additionally, we added SCM ultrasound capture photos for the longitude plane.
Page. 4. Line. 124. (AP) Please include the full terms before using an abbreviation.
Response: We appreciate your advice. Sorry for the omission of the full terms before using an abbreviation. We have corrected it again.
Page. 4. Line. 133. I think the figure in Figure 4 is too simple compared to the content mentioned in the text. It would be better to include a picture related to the diagnosis which was based on the findings.
Response: The wrong part of the picture was identified and corrected. And, we’ve added indications and dictionaries to help readers understand them more easily when they see them.
Page. 4. Line. 142. Please check the space between words.
Response: We have corrected it again.
Result Page. 5. Line. 150. Please put "children" after the number.
Response: We have corrected it again.
Page. 5. Line. 146. Please control the position of numbers and parentheses equally throughout the text (abstract, result, and discussion).
- ex) 80 boys (61.5%) or 45 (34.6%) children
Response: The positions of numbers and parentheses were modified equally within the full text.
Page. 5. Line. 156. (Table 2) Tables should be inserted into the main text close to their first citation.
Response: We appreciate your advice. Table 2 was relocated to the main text close to the first citation.
Page. 6. Line. 171. (Table 3) The number of patients in the normal pelvis group (n =79) and the number in the factor (delivery type, n = 81) do not match.
Response: Once again, we reviewed raw data and founded that we filled in the number of normal delivery groups incorrectly. We have corrected it again.
Discussion
It is very extensive and must be more specific.
1) Why were there no significant differences in the results?
Response: We expected that children with the mass type CMT would develop more PMS than children with non mass type CMT, but the result was different from our expectations. There was no difference between the mass type CMT and non mass type CMT in developing PMS.
The mass type CMT is thought to have a diverse clinical course and the outcome is depending on the histological characteristics of the fibrous mass. Generally, the CMT is diagnosed not at the base of muscle biopsy but the base of clinical feature and non-invasive ultrasonography, so that the histological feature could not be assessed precisely. Although the mass is not presented initially, the reduced muscle elasticity resulted in postural imbalance finally. Consequently, it is judged that the presence or absence of fibrotic mass did not affect the occurrence of the PMS.
The second point is the age of enrolled patients. This study enrolled patients who visited the clinic before 4 years old. According to standard practice guidelines for CMT, a cervical spine radiograph was taken immediately after visiting, and then every one year or two-year as a follow-up. Patients who visited the clinic under one year old began to show symptoms after about three years. However, not all children were equally taken radiogrphs at a particular time. We also believe that age is one of the variables affecting the results, and we will add the limitation of the study.
2) What are the important findings of this study?
Response: The key point of this study is that the decreased elasticity of SCM muscle by the
initial muscle injury can cause trunk posture asymmetry after the long term regardless of the
presence of the fibrotic mass. We want to communicate that it is necessary to do the long-term
regular follow-up to prevent PMS after being diagnosed with CMT. We add this emphasis in
the discussion section.
The grammar of the writing has some errors throughout the manuscript and needs to be checked and corrected. This will improve the work greatly and will make it more readable.
Response: To improve readability, we reviewed and corrected the wrong grammar errors once again.

Round 2
Reviewer 1 Report
This is a very interesting topic and needs to be explored and presented to the readership. However, the connection between the occurrence of lower extremity/pelvic malalignment and the occurrence of congenital muscular torticollis (CMT) needs to be elaborated and discussed. The cited sources actually address cervicothoracic scoliosis and/or cervical spine deformities accompanying CMT. Has anyone looked further down the spine at thoracolumbar scoliosis which makes more sense relative to lower spinal malalignment. Was there any correlation with the occurrence of hip dysplasia? The incidence of PMS that was found appears to be greater than that expected of infants with CMT but it would be interesting to explore any potential correlation.
The first reviewer mentioned that the emphasis on pain and cosmesis is unusual - there are many other issues that would potentially accompany scoliosis that would appear to be more troubling. In response to the prior reviewer's comments, the authors responded that pain was mentioned in response to requested activities but that still isn't addressed in the manuscript.
There are multiple places where language is awkward and/or improper leading to confusion to the reader's comprehension.
For example, in the abstract rather than "reviewed as the method of retrospective study" consider "reviewed retrospectively"
p. 2 "there were some children seen functional scoliosis" - ? meaning?
p. 2 "no study for regarding the development of PMS through long-term follow-up of children with CMT." Should the word "exists" be placed after the word "study"?
p. 3 "Also, children who got the surgical treatment for the remaining the fibrotic mass were excluded from this study" - correct?
p. 3 - omit "the" prior to "regular physical therapy" in the paragraph under the chart
p. 5 "Taking aAn initial cervical spine radiograph was taken to for ruleing out the congenital spine anomaly."
p. 6 When the authors talk about the "pelvic leveling at a standing position" - this is confusing. It seems that the authors mean "with the pelvis level in a standing position" - however, here and later, it turns out that the pelvis is not always level. This needs to be reworded as it does in the paragraph after the photos of the calipers at the child's pelvis where it says, "If the pelvis is leveled unequally..." -leveled suggests at the same height - perhaps better terminology would be "If the pelvis aligns unequally..."
p. 9 "Although not all children were taken radiograph at the same particular time" needs rewording to read "Although not all children had radiographs taken at the same timeline" or "at the same ages"
p. 11 Please use people first language - rather than "CMTTchildren" use "children with a history of CMT"
p. 11 "...be more likely to have PMS lately..." word should be "later"
p. 11 "It was inferred that the outcome different from the expectation..." insert "being" between "outcome" and "different"
p. 11 "...diagnosed not at the base...but the base of..." Both occurrences of "base" should be "basis"
p. 11 "Although the mass is not presented initially..." "presented" should be "present"
p. 11 "...resulted in postural imbalance finally." replace "finally" with "later"
p. 12 "The fine muscular fibrosis could be continued in the non-mass type so that..." consider replacing the word "fine" - meaning is unclear; also "could be continued" should be replaced by "may persist"
p. 12 "...diagnosed not at the base of muscle biopsy but the base of..." Both instances of the word "base" should be "basis
p. 12 "Although the mass is not presented initially, the reduced muscle elasticity resulted in postural imbalance finally." Replace "presented" with "present" and delete "finally" instead saying "reduced muscle elasticity eventually resulted..." or something similar
Author Response
Response to Reviewer Comments
We have sincerely appreciated your comments on improving the quality of our paper. We have addressed all the comments and made a major revision of our manuscript. Our responses to all your comments and the revisions are listed below.
This is a very interesting topic and needs to be explored and presented to the readership.
However, the connection between the occurrence of lower extremity/pelvic malalignment and the occurrence of congenital muscular torticollis (CMT) needs to be elaborated and discussed.
The cited sources address cervicothoracic scoliosis and/or cervical spine deformities accompanying CMT.
Has anyone looked further down the spine at thoracolumbar scoliosis which makes more sense relative to lower spinal malalignment? Was there any correlation with the occurrence of hip dysplasia? The incidence of PMS that was found appears to be greater than that expected of infants with CMT, but it would be interesting to explore any potential correlation.
Response: To best of our knowledge, there has been no study of the correlation between CMT and lower spinal malalignment, but we added the reference of association between the adolescent idiopathic scoliosis and pelvic obliquity.
[Reference] Line 387-388
- Cho, J.H.; Lee, C.S.; Joo, Y.; Park, J.; Hwang, C.J.; Lee, D. Association between Sacral Slanting and Adjacent Structures in Patients with Adolescent Idiopathic Scoliosis. Clinics in Orthopedic Surgery 2017, 9, 57-62.
On the other hand, the correlation of CMT and developmental dysplasia of the hip (DDH) has been established. Multiple studies reported a correlation between CMT and DDH at a rate between 2% and 29% [1]. According to the standard practice guideline for CMT of our institution, ultrasonography for screening of DDH was also being performed simultaneously at the initial evaluation. Alpha & beta angle of the hip joint were measured with ultrasonography. DDH was identified in children with CMT at a typical rate, although this study did not address it. Also, the case of CMT combined with DDH were excluded as the category of comorbid musculoskeletal problems.
We added the related it to the exclusion criteria of the subject.
[Added paragraph in method] Line 81-86
The exclusion criteria were as follows: (a) congenital anomalies of the spine, (b) spasmodic torticollis, (c) premorbid or comorbid musculoskeletal problems that affect the cervical range of motion, and (d) other congenital musculoskeletal anomalies of a lower extremity such as leg length discrepancy or hip dysplasia. Also, children who got the surgical treatment for the fibrotic mass were excluded from this study.
As mentioned in the discussion part, CMT decreased muscle elasticity resulting in differences in the active ROM of the neck. If this phenomenon is maintained for a long time without proper intervention, it can cause an asymmetric face and scoliosis. An altered cervical spine curve might provide a compensatory effect on the whole spine and pelvis in the coronal plane. Through this process, it is thought that it eventually develops into PMS.
The first reviewer mentioned that the emphasis on pain and cosmesis is unusual - there are many other issues that would potentially accompany scoliosis that would appear to be more troubling.
In response to the prior reviewer's comments, the authors responded that pain was mentioned in response to requested activities but that still isn't addressed in the manuscript.
Response: The first reviewer’s question was whether screening the spine x-ray was routine or for complaining of pain. Our team had many clinical data about children with muscular torticollis and found that they had gradual asymmetry of facial and spine curvature after a long period even if they had a mild initial symptom. Based on these clinical experiences, our institute has done a routine screening spine x-ray exam for children with CMT. Also, we reviewed the medical records of 130 children again, and there were no records related to the pain. The reason we mentioned the pain in our manuscript was that the children with PMS were more likely to complain about the pain at doing exercise.
We added an explanation for pain that we mentioned in the discussion.
[Added paragraph at Discussion] Line 279-280
Although these findings could increase the chances to complain of pain at doing exercise in children with CMT combined with PMS, there were no reporting pains in this study.
There are multiple places where language is awkward and/or improper leading to confusion to the reader's comprehension.
Response: We appreciate your advice. To improve readability, we reviewed and corrected the wrong grammar errors once again.
For example, in the abstract rather than "reviewed as the method of retrospective study" consider "reviewed retrospectively"
Response: Thank you for your comment. We corrected it.
- 2 "there were some children seen functional scoliosis" - ? meaning?
Response: Thank you for your comment. Functional scoliosis means that the curve is not fixed. It can be corrected by treating the underlying condition such as trauma, leg length discrepancy. Originally, we wrote the word “functional” because the spinal curve of scoliosis was not severe in our almost cases. But it gives the reader the other confusion. So, we change the sentence with secondary scoliosis.
[Changed paragraph at Discussion] Line 51-52
There were some children seen both secondary scoliosis and pelvic asymmetry among the children with a history of CMT.
- 2 "no study regarding the development of PMS through long-term follow-up of children with CMT." Should the word "exists" be placed after the word "study"?
Response: Thank you for your comment. We corrected it.
- 3 "Also, children who got the surgical treatment for the remaining fibrotic mass were excluded from this study" - correct?
Response: Thank you for your comment. We added the sentence.
- 3 - omit "the" prior to "regular physical therapy" in the paragraph under the chart
Response: Thank you for your comment. We changed it.
- 5 "Taking aAn initial cervical spine radiograph was taken tofor ruleing out the congenital spine anomaly."
Response: Thank you for your comment. We corrected it.
- 6 When the authors talk about the "pelvic leveling at a standing position" - this is confusing. It seems that the authors mean "with the pelvis level in a standing position" - however, here and later, it turns out that the pelvis is not always level. This needs to be reworded as it does in the paragraph after the photos of the calipers at the child's pelvis where it says, "If the pelvis is leveled unequally..." -leveled suggests at the same height - perhaps better terminology would be "If the pelvis aligns unequally..."
Response: Thank you for your comment. We corrected it.
- 9 "Although not all children were taken radiograph at the same particular time" needs rewording to read "Although not all children had radiographs taken at the same timeline" or "at the same ages"
Response: Thank you for your comment. We corrected it.
- 11 Please use people first language - rather than "CMT children" use "children with a history of CMT"
Response: Thank you for your comment. We corrected it.
- 11 "...be more likely to have PMS lately..." word should be "later"
Response: Thank you for your comment. We corrected it.
- 11 "It was inferred that the outcome different from the expectation..." insert "being" between "outcome" and "different"
Response: Thank you for your comment. We corrected it.
- 11 "...diagnosed not at the base...but the base of..." Both occurrences of "base" should be "basis"
Response: Thank you for your comment. We corrected it.
- 11 "Although the mass is not presented initially..." "presented" should be "present"
Response: Thank you for your comment. We corrected it.
- 11 "...resulted in postural imbalance finally." replace "finally" with "later"
Response: Thank you for your comment. We corrected it.
- 12 "The fine muscular fibrosis could be continued in the non-mass type so that..." consider replacing the word "fine" - meaning is unclear; also "could be continued" should be replaced by "may persist"
Response: Thank you for your comment. We corrected it. We change the word “ fine” to “slight”
- 12 "...diagnosed not at the base of muscle biopsy but the base of..." Both instances of the word "base" should be "basis
Response: Thank you for your comment. We corrected it.
- 12 "Although the mass is not presented initially, the reduced muscle elasticity resulted in postural imbalance finally." Replace "presented" with "present" and delete "finally" instead saying "reduced muscle elasticity eventually resulted..." or something similar
Response: Thank you for your comment. We corrected it.
